# A Gas Sensor Based on Network Nanowire for H$_2$S Monitor in Construction Waste Landfill

Pengyu Ren [1,2], Qingwei Shi [2,*] and Lingling Qi [2,*]

1    College of Architecture and Urban Planning, Chongqing Jiaotong University, Chongqing 400074, China; pengyu_ren@cqu.edu.cn
2    School of Management Science and Real Estate, Chongqing University, Chongqing 400045, China
*    Correspondence: shiqw@cqu.edu.cn (Q.S.); lynn.l.qi@cqu.edu.cn (L.Q.)

**Abstract:** As an extremely harmful gas, H$_2$S gas is the major pollutant in construction waste landfill. Herein, a one-dimensional oxide nanomaterial was produced from a simple wet chemical method to serve as a H$_2$S gas sensing material. The SEM observation indicates that the nanomaterial with network structure is constructed by a lot of nanowires with an approximate diameter from 24 nm to 40 nm. The sensing film was formed on a ceramic substrate using a slurry composed of the as-prepared network nanowires. Furthermore, a gas sensing measurement was carried out to determine the gas sensing performances towards the H$_2$S gas. The detection results at different working temperature towards various gas concentrations demonstrate that the network nanowires-based sensor exhibits a higher gas response to H$_2$S as compared to that of the rod-like one. The optimum working temperature of the network and rod-like nanomaterials is both 300 °C, and the corresponding maximum gas response is 24.4 and 13.6, respectively. Namely, the gas response of the network-based gas sensor is almost larger than that of the rod-like oxide. Moreover, the network nanowires-based gas sensor display a faster gas response and recovery speed. In addition, the fabricated gas sensors all exhibit excellent repeatability. Such improved sensing properties may offer a promising potential to realize an efficient detection of harmful H$_2$S gas released from construction waste landfill.

**Keywords:** network nanowires; nanorod; construction waste landfill; H$_2$S gas





## 1. Introduction

Detecting harmful and toxic gas is now considered to be significant around the world. Hydrogen sulfide (H$_2$S) especially is an extremely harmful gas with the smell of rotten eggs, which can be easily released from the construction waste landfill [1–3]. At present, construction waste is mostly treated via landfill. In the process of construction waste stacking, however, some organic substances decompose and then produce harmful gas under the function of temperature and moisture. For instance, construction waste gypsum contains a lot of sulfate ions, which will be converted into H$_2$S gas as in the following reaction.

$$2(CH_2O\text{-}R) + SO_4{}^{2-} \rightarrow 2HCO_3{}^- + H_2S + 2R \tag{1}$$

$$S^{2-} + H^+ \rightarrow HS^- \tag{2}$$

$$HS^- + H^+ \rightarrow H_2S \tag{3}$$

Therefore, the monitor of H$_2$S gas from construction waste landfill has received great attention due to its flammable and corrosive characteristics [4–6]. Nowadays, H$_2$S gas isdetected using a chemiresistor gas sensor based on metal oxide semiconductors because of its feasibility, which meets the demand of efficient gas detection [7–9].

There are a lot of metal oxide semiconductor-based gas sensors, such as n-type SnO$_2$, ZnO, WO$_3$, and p-type NiO, CuO [10–14]. Among of them, WO$_3$ is demonstrated as an effective oxide material for H$_2$S detection. In general, to enhance the gas sensing properties

of the $WO_3$-based gas sensors, modification of microstructures of $WO_3$ sensing material is deemed practicable. Nanostructures with a one-dimensional structure such as nanowires, nanofibers and nanorods have been widely studied owing to their good conductivity and higher surface area [15–17]. Kim et al. synthesized Si-coated $SnO_2$ nanowires as the sensitive material for the fabrication of $H_2S$ gas sensor [18]. Zhang et al. prepared ZnO-carbon nanofibers by an electrospinning route and a subsequent annealing treatment. The results showed that the $H_2S$ gas sensor exhibited an excellent response as compared to the pure one [19]. In conclusion, the construction of a one-dimensional nanostructure has vital performances resulting in desirable gas sensors because gas molecules have more contacting area and can effectively diffuse into or out of the sensing material, increasing the gas response and performing with a relatively fast response and recovery speed.

Herein, to evaluate the sensor response of the gas sensor, the surface characteristics of gas-sensing material could be tailored via hybrid structures. This work reports a simple preparation, characterization and $H_2S$ gas sensing application of $WO_3$ nanostructure-based sensors. To compare the sensing performances of the prepared network nanowires, the rod-like structure was also synthesized and examined. The as-prepared network $WO_3·H_2O$ is to make up fora large amount of nanowires intertwined with each other. Moreover, these nanowires construct a porous structure. The gas sensing measurement to $H_2S$ gas reveals that, as compared to the rod-like sample, the network nanowires-based gas sensor exhibits a higher sensor response of 24.4 when the sensor working temperature is 300 °C and the gas concentration of $H_2S$ is 50 ppm. In addition, the gas sensor made of the network $WO_3·H_2O$ nanowires display a quick response and recovery speed, and a good repeatability to $H_2S$ gas. The improvement mechanism is mainly attributed to the efficient surface area as well as the pore structure.

## 2. Experiments

### 2.1. Synthesis of the Network Nanowire Material

All of the chemical reagents were directly used in the process of sample preparation. $WO_3·H_2O$ nanowires were synthesized though blending sodium tungstate and acid as the raw materials in deionized water using a simple one-step wet chemical method (hydrothermal route). First, 0.49 gof sodium tungstate ($Na_2WO_3$) was dispersed in 30 mL deionized water with magnetic stirring. Then, 3 mol/L HCl was added into the solution until the pH value reached about 2, resulting in the precipitate formation. Thereafter, 0.38 g of oxalic acid ($H_2C_2O_4$) was added into the above solution with magnetic stirring until a uniform solution was formed. The obtained homogenous solution was transferred into a 50 mL Teflon-lined stainless steel autoclave and heated at 180 °C for 12 h. Then, the autoclave was cooled to room temperature in air. The precipitates were collected by high speed centrifugation (at 5000 rpm for 3 min) via washing with deionized water and ethanol three times. Afterwards, to obtain the final product, the precipitates were dried at 60 °C for 12 h in an oven. For comparison, the rod-like sample was synthesized via an analogous route of hydrothermal method and 0.45 g of $Na_2WO_3$ and 20 g $K_2SO_4$ were added into 30 mL deionized water under stirring. Next, 3 mol/L HCl was added into the solution until the pH value reached about 2. The obtained solution was sealed into the autoclave and kept at 120 °C for 10 h. After washing and centrifugation, the sample was dried at 60 °C for 12 h.

### 2.2. Sample Analysis

The crystal phase of the as-synthesized two samples was recorded by X-ray diffraction (XRD; D/MAX-1200X; Riguku International Co., Tokyo, Japan) using Cu-K$\alpha$ radiation at $2\theta$ from 10° to 70° with a step of 0.02°. The morphological features of the obtained two $WO_3·H_2O$ samples were analyzed through filed emission scanning electron microscopy (FE-SEM; SU8020; Hitachi Ltd., Tokyo, Japan) operating at 3.0 kV. The gas sensing performances to $H_2S$ gas were detected using an intelligent gas sensor system.

### 2.3. Sensor Fabrication

The indirect-heated gas sensor was fabricated as the following step: a certain amount of the as-prepared $WO_3 \cdot H_2O$ powder was ultrasonically dispersed into deionized water to form a suspension. The obtained suspension was then coated onto an alumina ceramic tube to form uniforma thin sensing film. The thickness of the sensing film was determined to be approximately 100 μm analyzed by optical microscope. Such an alumina tube was served as the substrate to support the sensing material. In particularly, the alumina tube was very small, only 4 mm in length and 1 mm in outer diameter. A pair of gold rings as electrodes and four platinum wires as conducting wires were fixed on both sides of the alumina tube beforehand (as shown in Figure 1a). After the coated sensing film was driednaturally, a Ni-Cr alloy wire was inserted into the treated alumina tube. Finally, the alumina tube was welded on a black pedestal through soldering. Before measurement, the as-fabricated gas sensor was aged at 250 °C for several days in order to enhance its stability and repeatability.

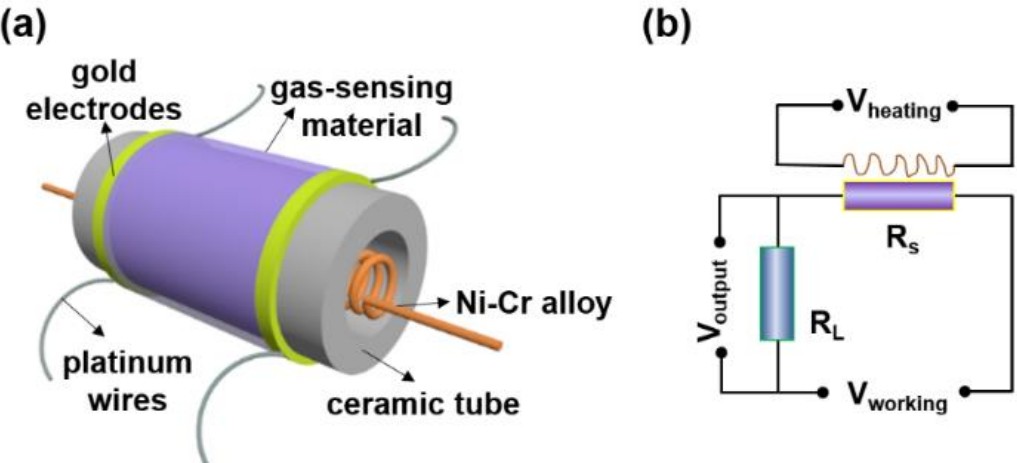

**Figure 1.** (**a**) The diagrammatic sketch of side-heated gas sensor. (**b**) The electric circuit of thegas sensing test.

### 2.4. Sensor Measurement

After the successful fabrication of the gas sensor was made of the $WO_3 \cdot H_2O$ samples, the gas response of the as-fabricated gas sensors was determined using the intelligent gas sensing system. The electric circuit of gas sensing detection is shown in Figure 1b. A load resistor ($R_l$) is connected in series with the as-fabricated gas sensor with a certain voltage ($V_{working}$). The resistance value of the gas sensor ($R_s$) can be calculated by measuring the output voltage of the load resistance ($V_{output}$). The operating temperature of the gas sensor is automatically controlled by changing the heating voltage ($V_{heating}$). The sensor resistance is varied when a target gas is introduced into the test chamber, leading to the simultaneous change of the load resistor. In this work, the gas response of the gas sensor is defined as follows: Gas response $S = R_a / R_g$. $R_g$ is the resistance value of gas sensor in target gas, and $R_a$ represents the resistance value of gas sensor in air. Additionally, the response and recovery time is usually expressed as follows: the response time is the time required for the sensor resistance to achieve 90% of the steady resistance state from its initial resistance state. The recovery time is the time to achieve 90% of total resistance change after the target gas is removed. The target gas is the standard gas and is utilized in a dry form.

## 3. Results and Discussion

### 3.1. Structural Information

The purity and crystal structures of the hydrothermally synthesized $WO_3 \cdot H_2O$ samples were evaluated by X-ray diffraction (XRD) analysis. Figure 2 presents the XRD patterns of the prepared $WO_3 \cdot H_2O$ samples. The diffraction peaks of both $WO_3 \cdot H_2O$ samples at

16.5°, 25.6°, 34.9°, and 56.7°, matched with the (0 2 0), (1 1 1), and (0 0 2) crystal planes, which are well indexed to a hexagonal structure of $WO_3 \cdot H_2O$ when compare with the No. 84-0886 of Jiont Committee on Powder Diffraction Standards card (JCPDS). The sharp and intense diffraction peaks demonstrate the high crystallinity of the $WO_3 \cdot H_2O$ samples. Moreover, no other diffraction peaks originating from impurities are clearly detected, indicating a relatively high-purity of the two kinds of $WO_3 \cdot H_2O$ samples. The EDS energy spectra is displayed in Figure 2b, which indicates the presence of W and O elements. This further confirms the successful preparation of the tungsten oxide sample.

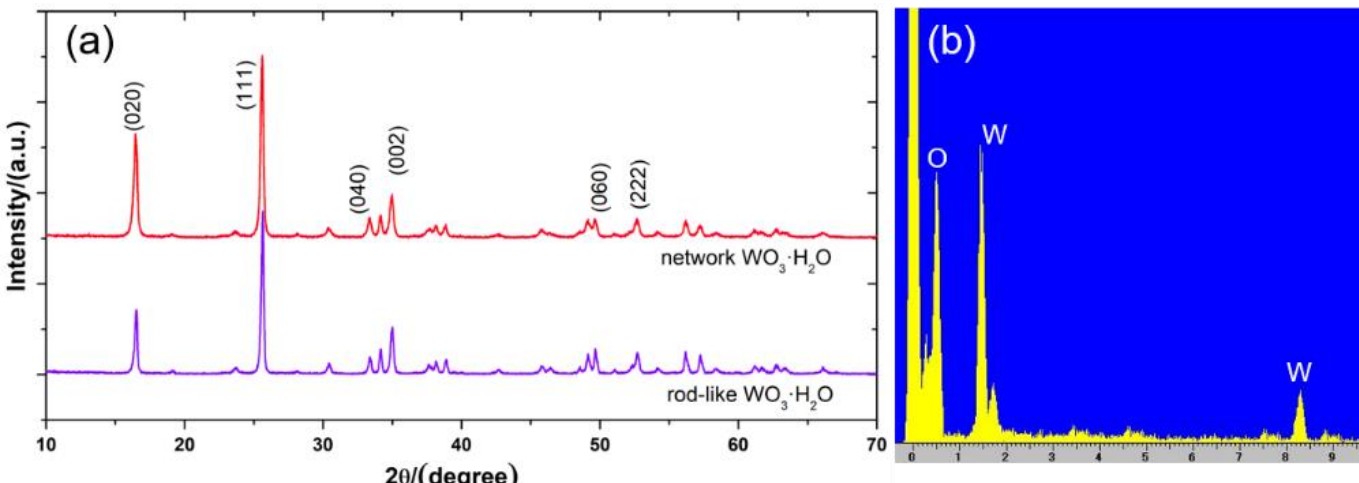

**Figure 2.** (**a**) XRD patterns of the as-prepared rod-like and network nanowire samples. (**b**) EDS of the network nanowire sample.

To analyze the microstructure of the as-prepared $WO_3 \cdot H_2O$ samples, the SEM was performed. The obtained morphological images with various magnifications are displayed in Figure 3. From Figure 3a–c, it can be clearly seen that this $WO_3 \cdot H_2O$ sample is composed of a large number of rod-like structures with irregular shapes. The average diameter of the rod is about 65 to 95 nm. A lot of nanoparticles are mixed in them. It is worth noting that these rod-structures are stacked tightly together. Additionally, the microstructure of another sample is showed in Figure 3d–f. It can be found that this $WO_3 \cdot H_2O$ sample is analogous to a flat silk fabric, which has a network structure and a large number of pores (as shown in Figure 3d). Moreover, it can be seen from the enlarged Figure 3e that the flat silk fabric is composed of numerous one-dimensional nanowires. These nanowires are interweaved with each other, constructing abundant pore structures. The diameter of the nanowires can be analyzed from the further enlarged Figure 3f. The diameter of these nanowires is about 24 to 40 nm. Incomparison, the structure of the as-prepared one-dimensional network $WO_3 \cdot H_2O$ nanowires is beneficial for gas sensing interaction because more contacting area and diffusion pathways are provided. The $N_2$ adsorption–desorption isotherm was carried out, and the results are shown in Table 1. The pore size distribution for the nanorod and network nanowire is around 11.6 and 15.2 nm, respectively. These porous structures boost the diffusion of gas molecules. In addition, the network nanowire has a higher specific surface area of 41.3 $m^2 \ g^{-1}$ rather than that of the nanorod (25.4 $m^2 \ g^{-1}$). The higher specific surface area increases the contacting area between oxygen ions and target gas.

**Table 1.** BET surface area and pore size distribution of the rod-like and network nanowire sample from $N_2$ adsorption and desorption isotherms.

| Sample | BET Surface Area ($m^2 \ g^{-1}$) | Pore Size (nm) |
|---|---|---|
| Rod-like sample | 25.4 | 11.6 |
| Network nanowire sample | 41.3 | 15.2 |

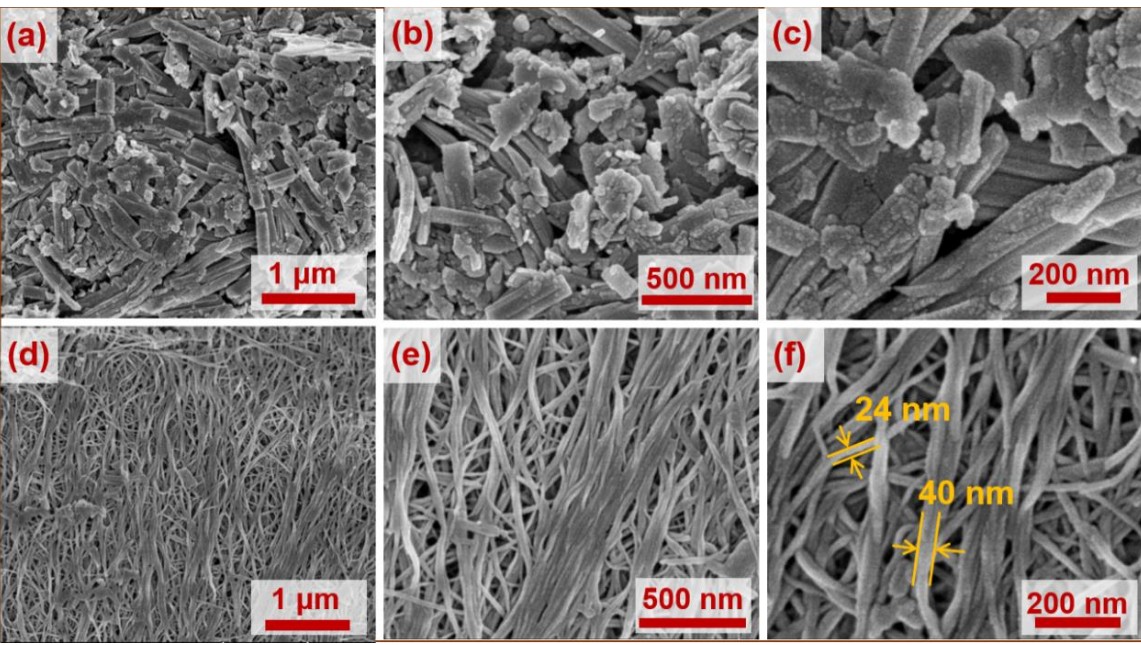

**Figure 3.** SEM images of the as-prepared (**a**–**c**) rod-like and (**d**–**f**) network $WO_3 \cdot H_2O$ material with different magnification.

### 3.2. Gas Sensing Analysis

The gas sensitive properties of the $WO_3 \cdot H_2O$ samples were systematically investigated. As is known to us, the working temperature of the metal-oxide semiconductor-based gas sensors is of great importance to affect the gas sensing characteristics. First, the as-fabricated two gas sensors using the hydrothermally synthesized $WO_3 \cdot H_2O$ samples are operated at changed temperatures to determine an impressive gas response in the range of 100–450 °C. As displayed in Figure 4a, the two gas response patterns to different working temperatures show thesame trend of first an increase and then a decrease. Obviously, the gas response of the network $WO_3 \cdot H_2O$ is higher than that of the rod-like one. The maximum gas responses of the network and rod-like $WO_3 \cdot H_2O$ are 24.4 and 13.6 at a certain working temperature of 300 °C, respectively. It can be seen that the gas response of the network nanowire is almost twice as large as the rod-like structure. When the gas sensors are operated at relatively low temperatures, the thermal energy could not conquer the activation energy barrier, leading to aninadequate interaction with the sensing materials. However, when the operating temperature rises, higher thermal energy will be provided for the gas molecules to overcome the activation energy barrier. Therefore, the sensing interaction between gas molecules and sensitive materials is promoted, and the ion adsorption process on the surface of sensitive material is improved. However, when the working temperature continues to increase and exceed the optimum working temperature, the adsorbed oxygen molecules will get adequate energy to divorce from the sensing material surface before the sensing reaction happens. Consequently, the gas response value of the sensor will be decreased. As shown in Figure 4a, the optimum working temperature of the both gas sensors for 50 ppm $H_2S$ gas is 300 °C. As can be seen in Figure 4b, with the $H_2S$ gas concentration increase, the gas responses show an increased trend of linear growth in the tested gas concentration range (from 10 to 80 ppm). However, it is apparent that the gas response of the network nanowire-based gas sensor is larger than that of the rod-like structure-based sensor. After the linear fitting, the response curve of the network $WO_3 \cdot H_2O$ nanowire-based gas sensor as afunction of gas concentration can be described as $S = 0.55179C - 2.71786$ ($R^2 = 0.99961$). The other one can be described as $S = 0.31333C - 2.125$ ($R^2 = 0.99905$). The $R^2$ is the fitting degree. $R^2$ values in this work are very close to 1.0, indicating a high linearity. In addition, the LOD for the nanorods and the network nanowires are 6.78 and 4.93 ppm, respectively. A lower LOD value for the network nanowires indicates its effective detectionin practical application.

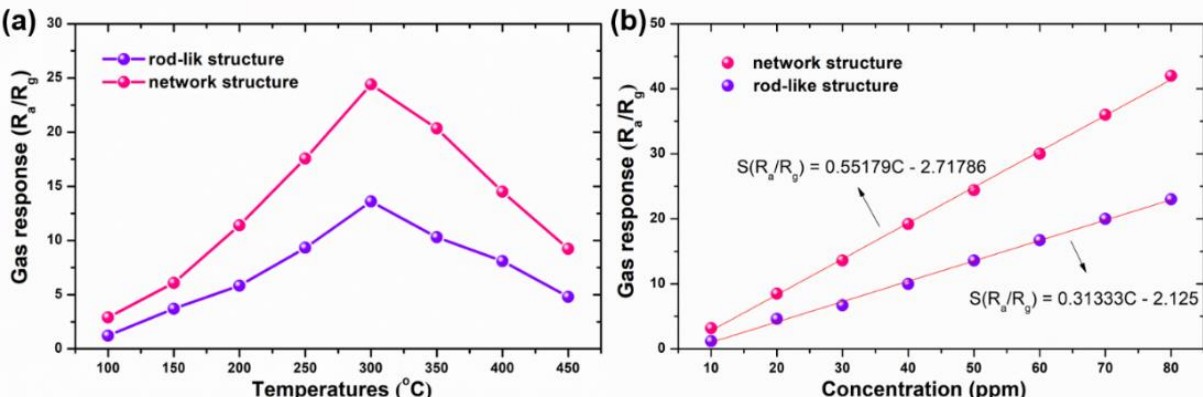

**Figure 4.** (**a**) The gas response of the network and rod-like WO$_3$·H$_2$O-based gas sensors at different operating temperatures between 100 °C to 450 °C to 50 ppm H$_2$S. (**b**) The gas response of the network and rod-like WO$_3$·H$_2$O-based gas sensors to different gas concentration between 10 ppm to 80 ppm at 300 °C.

The dynamic response and recovery curves of the network and rod-like WO$_3$·H$_2$O-based gas sensors towards 50 ppm H$_2$S at 300 °C are shown in Figure 5a. When the reduced target gas is introduced into the test chamber, the gas responses for both gas sensors are rapidly increased due to the gas sensing reaction. Once the target gas gets out, the gas responses for both gas sensors are decreased and reach their initial states. However, there is a difference in their rate of response change. For the network WO$_3$·H$_2$O gas sensor, the response and recovery times are evaluated to be 6 s and 16 s, respectively. For the rod-like WO$_3$·H$_2$O gas sensor, the response and recovery times are evaluated to be 9 s and 18 s, respectively. Incomparison, the network WO$_3$·H$_2$O nanowire-based gas sensor exhibits a faster response and recovery speed. Furthermore, good repeatability is instrumental in gas sensors in practical application. Therefore, the as-fabricated gas sensors based on the two WO$_3$·H$_2$O samples were tested for 5 cycles towards 50 ppm H$_2$S gas, and the response and recovery curve is displayed in Figure 5b. From the obtained results, it can be seen that both gas sensors demonstrate a similar response and recovery performance. Namely, the gas response is hardly changed towards 50 ppm H$_2$S gas at the operating temperature of 300 °C during the 5 cycles of test, which indicates that the two WO$_3$·H$_2$O-based gas sensors show an excellent repeatability. The selectivity tests were performed at 300 °C towards 50 ppm of H$_2$S, ethanol, H$_2$, CO, and NO. As shown in Figure 6, the network nanowire-based gas sensor exhibits a different gas response value to the tested target gas. The gas response of the sensor to H$_2$S gas is much higher than the that of other gases, thus showing the selectivity of the network nanowire-based sensor towards H$_2$S.

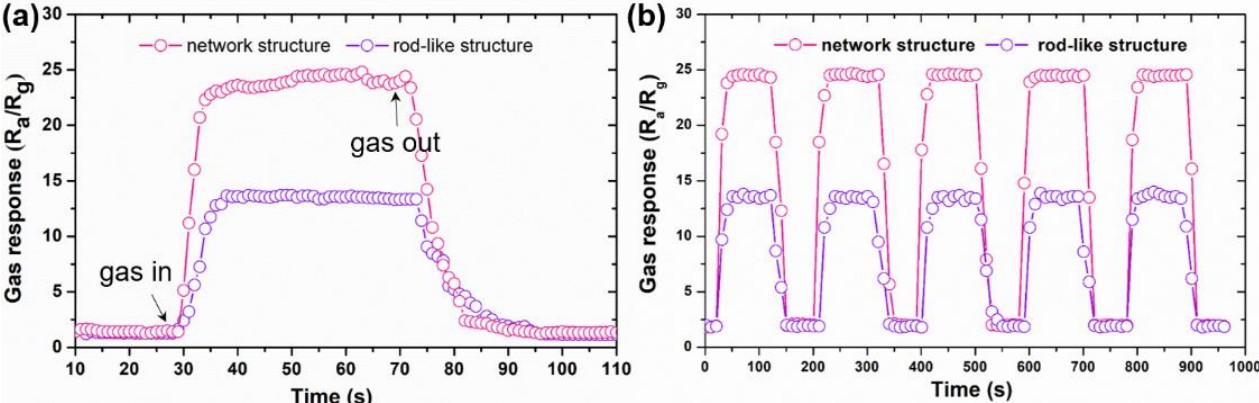

**Figure 5.** (**a**) The dynamic response and recovery curves of the network and rod-like WO$_3$·H$_2$O-based gas sensors towards 50 ppm H$_2$S at 300 °C. (**b**) The cycling response and recovery curves of the network and rod-like WO$_3$·H$_2$O-based gas sensors towards 50 ppm H$_2$S at 300 °C.

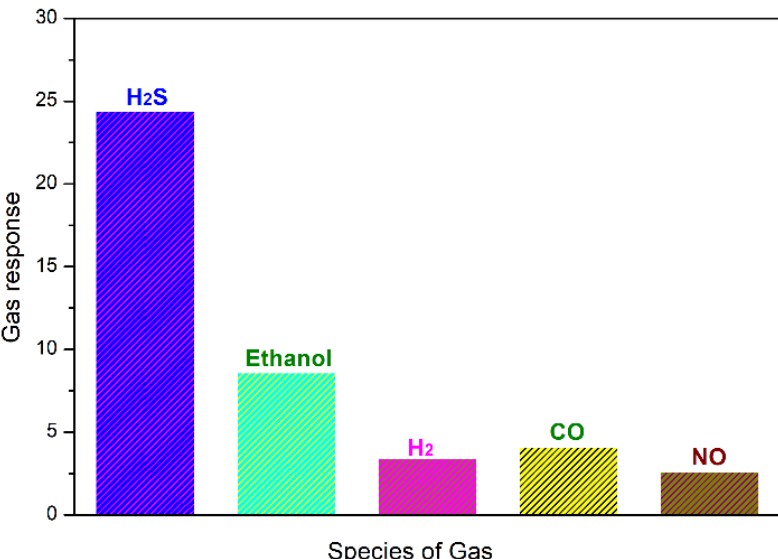

**Figure 6.** Selectivity of the network nanowire-based gas sensor to different gas at 50 ppm.

*3.3. Gas Sensing Mechanism*

It is well known that the gas sensing mechanism of metal oxide semiconductor-based gas sensors can be elucidated in the following way: the surface sensing reaction is caused by the resistance change owing to the adsorption and desorption of test gas molecules on the sensitive material surface, which is the main reason for the sensor response change [20,21]. As a result, explaining the gas sensing mechanism for $WO_3 \cdot H_2O$-based gas sensors is necessary. For $WO_3$, $\cdot H_2O$ as the typical n-type metal oxide semiconductor, the $WO_3 \cdot H_2O$-based gas sensor is first exposed in air and oxygen molecules closed to the sensing material will be easily adsorbed on the surface of the n-type $WO_3 \cdot H_2O$ material byway of physical adsorption or chemical adsorption. Owing to their high electron affinity of 0.43 eV, the adsorbed oxygen molecules will capture free electrons, which come from conduction bands of sensitive material, resulting in the formation of various adsorbed oxygen ions on the sensor surface, such as $O_2^-$, $O^-$ and $O^{2-}$. Thereafter, the consumption of free electrons leads to the formation of a electron depletion layer on the sensitive material surface, reducing the conductive channels of the sensitive $WO_3 \cdot H_2O$ material. Hence, the reduction of electron concentration in sensitive $WO_3 \cdot H_2O$ material contributes to a great increase insensor resistance. The abovementioned sensing reaction occurring on the surface of sensing material can be expressed as follows (Equations (4)–(7)):

$$O_2 \text{ (gas)} \rightarrow O_2 \text{ (ads)} \tag{4}$$

$$O_2 \text{ (ads)} + e^- \rightarrow O_2^- \text{ (ads)} \tag{5}$$

$$O_2^- \text{ (ads)} + e^- \rightarrow 2O^- \text{ (ads)} \tag{6}$$

$$O^- \text{ (ads)} + e^- \rightarrow O^{2-} \text{ (ads)} \tag{7}$$

When the as-synthesized $WO_3 \cdot H_2O$ sample-based gas sensor is introduced to the target gas, such as $H_2S$ gas, the $H_2S$ molecules will be adsorbed on the surface of sensitive material as well, and the $H_2S$ gas molecules will interact with the formed oxygen ions, leading to the generation of $H_2O$, $SO_2$, and free electrons. The sensitive reaction can be described as follows (Equation (8)):

$$2H_2S \text{ (gas)} + 3O^{2-} \text{ (ads)} \rightarrow 2H_2O + 2SO_2 + 3e^- \tag{8}$$

More free electrons will be released and come back to the conduction band of the prepared $WO_3 \cdot H_2O$ material due to the process of sensing reaction with target gas molecules.

Consequently, the thickness of the electron depletion region in the sensitive material will be reduced. Hence, in the $H_2S$ gas atmosphere, the resistance value of the $WO_3 \cdot H_2O$-based gas sensor will be decreased. Once the $WO_3 \cdot H_2O$-based gas sensor is put into fresh air again, the sensor resistance will be increase until the generation of new equilibrium. In this work, the as-synthesized network $WO_3 \cdot H_2O$ nanowire-based gas sensor presents enhanced gas response to $H_2S$ gas, the reason forwhich can be concluded as follows (Figure 7). On the one hand, the efficient surface area based on the analysis of surface morphology is beneficial for the surface sensing interaction, whichis considered animportant reason forthe excellent gas sensing performance of the fabricated gas sensor based on the network $WO_3 \cdot H_2O$ sample in this work. On the other hand, a large number of pore structures constructed by the interconnected nanowires in the network $WO_3 \cdot H_2O$ is conducive to the gas diffusion, resulting in afaster response and recovery speed.

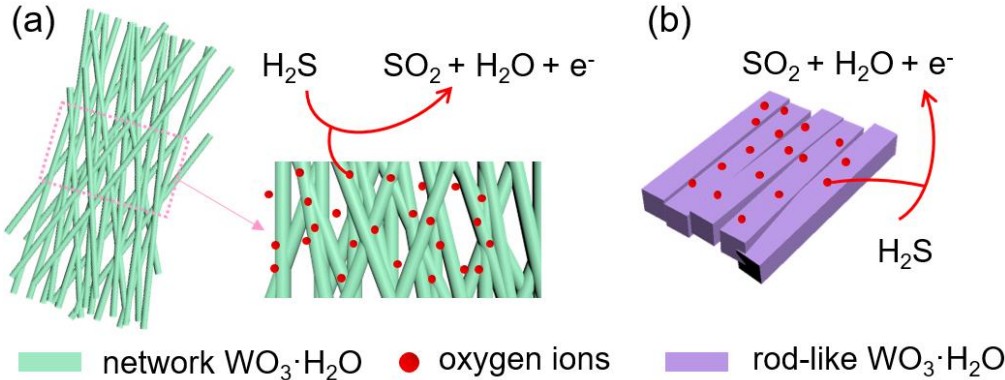

**Figure 7.** Gas sensing mechanism of the (**a**) network $WO_3 \cdot H_2O$ and (**b**) rod-like $WO_3 \cdot H_2O$.

## 4. Conclusions

In this study, an excellent gas sensing material for $H_2S$ gas measurement was successfully prepared using a simple one-step hydrothermal method. It is found that the network $WO_3 \cdot H_2O$ sample is composed of numerous nanowires. After gas sensing measurement towards different concentrations of $H_2S$ gas (10–80 ppm) at various operating temperatures (100–450 °C), the results indicate that network $WO_3 \cdot H_2O$ nanowire-based gas sensors possess an excellent gas response to $H_2S$ gas as compared to thoseof the rod-like structure, which may contribute to the network structure with numerous pores. The gas response value is evaluated to be 24.4 and 13.6 at the optimum working temperature of 300 °C for the network and rod-like structure, respectively. The testing results also reveal that the fabricated gas sensor exhibits good repeatability for 50 ppm $H_2S$ gas at the working temperature of 300 °C. Such enhanced gas sensing properties of the network $WO_3 \cdot H_2O$-nanowire-based gas sensor could pave a new way for the future development of an $H_2S$ monitor in construction waste landfill.

**Author Contributions:** Conceptualization, P.R. and L.Q.; investigation, P.R.; writing—original draft preparation, P.R.; writing—review and editing, Q.S. and L.Q.; All authors have read and agreed to the published version of the manuscript.

**Funding:** This research received no external funding.

**Institutional Review Board Statement:** Not applicable.

**Informed Consent Statement:** Not applicable.

**Data Availability Statement:** Not applicable.

**Conflicts of Interest:** The authors declare no conflict of interest.

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
