# Peer review of "A Gas Sensor Based on Network Nanowire for H2S Monitor in Construction Waste Landfill"

_chemosensors, doi:10.3390/chemosensors9070156_

Round 1

Reviewer 1 Report

The manuscript taking deal with improvement of H2S sensors to make them more sensitive.  The authors decide to modify the structure of W3O H2O sensor body. As a result, the authors present the nanowire “network” structure sensor with higher sensitivity, in compare with rod-like structure. Such research can be interesting for wide area of people who working in the field pollution reduction.

The paper was written in logical and clear form, however there are some points to be clarified.

Line 82. “For comparison, the rod-like sample was synthesized via an analogous route.” What was the difference in preparation? How to switch from nanowire to rod-like structure.  To my opinion it should be described or placed a reference to appropriate paper.

Line 126. XRD is insufficient for purity declaration. There is a possibility of contamination by organic and/or amorphous residuals. Such residuals would influence to material properties. Purity should be confirmed by EDS/WDS, XPS, SIMS or other methods.  

Line 146.  “It is worth noting that these rod-structure are stacked tightly together”. This statement is suggestive. The porosity seems to play key role in the material properties, so it should be evaluated.    

Line 209. Figure 5. It would be desirable to mark in the exact time of gas implementation. There is usually a delay before the beginning of response.  

Finally, the presented results (increasing of response for H2S) are interesting, however, the explanation of the fact is not solid without quantitative evaluation quantitatively and compare porosity ( or effective surfaces) for both types of samples.   

Author Response

Reviewer #1: The manuscript taking deal with improvement of H2S sensors to make them more sensitive. The authors decide to modify the structure of W3O·H2O sensor body. As a result, the authors present the nanowire “network” structure sensor with higher sensitivity, in compare with rod-like structure. Such research can be interesting for wide area of people who working in the field pollution reduction.

The paper was written in logical and clear form, however there are some points to be clarified.

Line 82. “For comparison, the rod-like sample was synthesized via an analogous route.” What was the difference in preparation? How to switch from nanowire to rod-like structure.  To my opinion it should be described or placed a reference to appropriate paper.

We appreciate the reviewer for this comment. We are sorry that we did not make the preparation clear. Similarly, the hydrothermal method was used to prepare the rod-like sample. The preparation process is described as following and also added in the revised manuscript. In this manuscript, in fact, the preparation relationship between nanowire and rod-like structure is not clear yet, the evolution mechanism will be further investigated in our future work.

For comparison, the rod-like sample was synthesized via an analogous route of hydrothermal method. 0.45 g of Na2WO3 and 20 g K2SO4 were added into 30 mL deionized water under stirring. And 3 mol/L HCl was added into the solution utile the pH value reached about 2. The obtained solution was sealed into the autoclave and kept at 120 oC for 10 h. After washing and centrifugation, the sample was dried at 60 oC for 12 h.

Line 126. XRD is insufficient for purity declaration. There is a possibility of contamination by organic and/or amorphous residuals. Such residuals would influence to material properties. Purity should be confirmed by EDS/WDS, XPS, SIMS or other methods.

We thank the reviewer for this comment. Exactly, the purity should be confirmed in this work. The EDS energy spectra was added as shown in Fig. 2b as follows.

The EDS energy spectra is displayed in Fig. 2b, which indicates the presence of W and O elements. This further confirms the successful preparation of tungsten oxide sample.

Line 146.  “It is worth noting that these rod-structure are stacked tightly together”. This statement is suggestive. The porosity seems to play key role in the material properties, so it should be evaluated. 

We thank the reviewer for this comment. The porosity and surface area should be evaluated in this work. And relevant discussion was added as follows.

The N2 adsorption-desorption isotherm was carried out, and the results are shown in Table 1. The pore size distribution for the nanorod and network nanowire is around 11. 6 and 15.2 nm, respectively. These porous structure boosts the diffusion of gas molecules. In addition, the network nanowire has higher specific surface area of 41. 3 m2 g-1 rather than that of nanorod (25.4 m2 g-1). The higher specific surface area increases the contacting area between oxygen ions and target gas.

Table 1 BET surface area and pore size distribution of the rod-like and network nanowire sample from N2 adsorption and desorption isotherms.

Sample

BET surface area (m2g-1)

Pore size (nm)

Rod-like sample

25.4

11.6

Network nanowire sample

41.3

15.2

Line 209. Figure 5. It would be desirable to mark in the exact time of gas implementation. There is usually a delay before the beginning of response. 

We thank the reviewer for this comment and agree that the reviewer point out an important issue. The relevant information has been added in the revised manuscript.

Finally, the presented results (increasing of response for H2S) are interesting, however, the explanation of the fact is not solid without quantitative evaluation quantitatively and compare porosity (or effective surfaces) for both types of samples.

We appreciate the reviewer for this comment. The explanation should be described quantitatively. The porosity and specific surface area have been performed in this work.

Reviewer 2 Report

In this paper, the authors fabricated novel metal oxide sensors for H2S gas with specific structures. The manuscript contains an original contribution to improve the sensitivity of a gas sensor for harmful gases. In addition, the new design of the gas sensor allows increasing the response performance for metal oxide sensors. However, the manuscript still has insufficient explanations of the sensor characteristics and results, as well as some grammatical mistakes. The authors should consider the comments in the attached file and revise the manuscript to improve the manuscript before publication in ‘Chemosensors’.

Author Response

Reviewer #2: In this paper, authors fabricated novel metal oxide sensor for H2S gas with specific structure. The manuscript contains an original contribution to improve the sensitivity of a gas sensor for harmful gases. In addition, the new design of the gas sensor allows increasing the response performance for metal oxide sensors. However, the manuscript still has insufficient explanation of the sensor characteristics and results, as well as some grammatical mistakes. The authors should consider the following comments and revise the manuscript to improve the manuscript before publication in ‘Chemosensors’.

  1. There are small mistakes in text mentioned following.

1-1   Page 1, equation (1), the elementary balance between left-hand side and right-hand side is not precise. Please revise equation (1).

1-2   Page 2, line 88, The formal name of FESEM is not correct; “filed scanning electron microscopy” should be “filed emission scanning electron microscopy”. Please revise the name of FESEM.

1-3   Page 4 Figure 2, “XRD spectrum ~” should be “XRD patterns ~” or “XRD curves ~”. “Spectrum” means the data of a measurement with frequency modulation of input or output signal. Please revise it.

1-4   There are some mistakes of “space”, page 2 line 67 “~ wassuccessively ~”, page 2 line 73 “0.49 gof ~”, page 3 line 133 “~ samples.Moreover, ~”, page 4 line 138 “~ WO3H2Osamples, ~”. Please check and revise it.

1-5   Page 4 line 149, “~ 24 nm to 40 nm.” should be “~ 24 to 40 nm.”. Please revise it.

Before resubmission, please check your manuscript again.

We appreciate the reviewer for this comment. And we have revised the English grammar and expression for our entire manuscript.

  1. In abstract, it is better to add some explanations to emphasize your original contributions, for example, adding “hydrothermal method” to explain “a simple wet chemical method”, and explain the reason why the performance of a network type sensor is better than rod type one.

As the reviewer suggested, we have provided the necessary information in abstract section as follows/

The improvement mechanism is mainly attributed to the efficient surface area as well as the pore structure.

  1. In section of Experiments, 2.3 Sensor Fabrication, please add the information of sensor characteristics, for example, the loading amount of sensor materials on the alumina tube substrate, how the authors decide the size of sensor, and the size of gold electrodes and Pt wires.

We thank the reviewer for this comment. Different thickness affects the sensing performances. As the reviewer suggested, we have provided the necessary information about the gas sensing film thickness for gas-sensing measurement and add following sentence. In addition, the ceramic tube, gold electrodes and Pt wires are all commercial, so their size are fixed.

The thickness of the sensing film was determined to be approximately 100 μm analyzed by optical microscope.

  1. In section of Results and discussion, about figure 3, please add the explanation of prepared samples in main manuscript, such as the average size and size distribution of network and rod type WO3·H2O materials, and its specific surface area, if possible.

We appreciate the reviewer for this comment. The average diameter of the rod type WO3·H2O is about 65 to 95 nm. The diameter of these nanowires of network structure is about 24 to 40 nm. The explanation of prepared samples has been added in the revised manuscript. Also, the specific surface area was added in the revised manuscript.

  1. In section of Results and discussion, about figure 4, are there any change of sensor response behavior depending on the temperature and any composition change after high temperature operation? In addition, when the concentration of H2S gas was changed, how did the ratio of the sensor response changed? If the authors have the results, please add it in the manuscript.

We thank the reviewer for this comment. The temperature is an important factor that affect the sensor response, as displayed in Figure 4a in the manuscript. The sensor response is changed depending on the operating temperature. the sensor response is increased firstly and then decreased with the operating temperature rising. In addition, the ratio of the sensor response is changed with the gas concentration increasing. Actually, the sensor response is approximately leveling off at higher gas concentration.

  1. In section of Results and discussion, about figure 5, are there any relationship between the surface area size and the resistance modulation of the sensor? In addition, how does the loading amount of the sensor materials affect sensor response?

We thank the reviewer for this comment. The porosity and surface area should be evaluated quantitatively in this work. In fact, higher specific surface area would result in larger gas response, because more active site could be provided for gas molecules. In addition, the loading amount of the sensing materials would affect the gas response. The optimized thickness of the sensing layer is existed. However, the thickness of the sensing film was determined to be approximately 100 μm in this work. the optimized thickness will be investigated in our future work. And relevant discussion was added as follows.

The N2 adsorption-desorption isotherm was carried out, and the results are shown in Table 1. The pore size distribution for the nanorod and network nanowire is around 11. 6 and 15.2 nm, respectively. These porous structure boosts the diffusion of gas molecules. In addition, the network nanowire has higher specific surface area of 41. 3 m2 g-1 rather than that of nanorod (25.4 m2 g-1). The higher specific surface area increases the contacting area between oxygen ions and target gas.

Table 1 BET surface area and pore size distribution of the rod-like and network nanowire sample from N2 adsorption and desorption isotherms.

Sample

BET surface area (m2g-1)

Pore size (nm)

Rod-like sample

25.4

11.6

Network nanowire sample

41.3

15.2

  1. In section of Results and discussion, does the authors detect produced SO2 gas from the sensor during sensor operation?

We appreciate the reviewer for this comment. SO2 could be produced due to the chemical reaction between H2S and oxygen ions. However, during the sensor operation, we did not detect the SO2 produced from the sensor.

Reviewer 3 Report

The authors synthesized nanowire and nanorod WO3 and investigated their H2S sensing characteristics. It was found that the nanowire WO3 with networked structure exhibited a higher response to H2S.

First of all, the authors must state clearly what is new in the study.

I could not figure out the novelty in this study.

Specific comments are as follows:

If the novelty lies in the synthesis process, the authors describe the novelty clearly, e. g., how the authors control the morphology of the WO3 nanostructure.

Concerning the sensing performance, the response is not so high compared with previous studies: e.g. ref. 15. (The authors must survey the related literature more thoroughly.)  Additionally, the concentration range tested is far satisfactory for landfill monitoring, which would need the detection limit of at least the order of ppm. The LOD should be also specified.

The authors are advised to provide the selectivity data.

As a whole, I have to judge the manuscript does not reach the criteria for publication.

Author Response

Reviewer #3: The authors synthesized nanowire and nanorod WO3 and investigated their H2S sensing characteristics. It was found that the nanowire WO3 with networked structure exhibited a higher response to H2S. First of all, the authors must state clearly what is new in the study. I could not figure out the novelty in this study.

Specific comments are as follows:

If the novelty lies in the synthesis process, the authors describe the novelty clearly, e. g., how the authors control the morphology of the WO3 nanostructure.

We appreciate the reviewer for this comment. We are sorry that the novelty was not described clearly. Herein, to evaluate the sensor response of gas sensor, the surface characteristics of gas-sensing material could be tailored via hybrid structures. This work reports a simple preparation, characterization and H2S gas sensing application of WO3 nanostructure based sensors. To compare the sensing performances of the prepared network nanowires, the rod-like structure was also synthesized and examined. The relevant discussion has been added in the revised manuscript as follows.

Concerning the sensing performance, the response is not so high compared with previous studies: e.g. ref. 15. (The authors must survey the related literature more thoroughly.)  Additionally, the concentration range tested is far satisfactory for landfill monitoring, which would need the detection limit of at least the order of ppm. The LOD should be also specified.

We thank the reviewer for this comment. The gas response was defined in a different way. In this manuscript, the gas response was defined as the ratio of resistance in air and in target gas (S = Ra/Rg), which was usually used in many other reports. But in Ref. 15, the gas response was defined as S = Ra/Rg ×100(%). In addition, the LOD for the nanorods and network nanowires are 6.78 and 4.93 ppm, respectively. lower LOD value for the network nanowires indicate its effective detecting in practical application.

The authors are advised to provide the selectivity data.

We thank the reviewer for this comment. The selectivity data has been added in the revised manuscript as follows.

The selectivity tests was performed at 300 oC towards 50 ppm of H2S, ethanol, H2, CO, NO. as shown in Figure 6, the network nanowire based gas sensor exhibits different gas reponse value to the tested target gas. Specially, the gas response of the sensor to H2S gas is much higher than the that of other gases, thus showing the selectivity of the network nanowire based sensor towards H2S.

Figure 6. Selectivity of the network nanowire based gas sensor to different gas at 50 ppm.

Round 2

Reviewer 1 Report

In my opinion, most of the ambiguities are now explained / corrected. Thanks to the authors for their efforts.  

Author Response

In my opinion, most of the ambiguities are now explained / corrected. Thanks to the authors for their efforts.  

Reply

We thank the reviewers for his/her uesful comment for your revised manuscript. We have touched up every sentences and corrected many spell or grammar mistakes. Now the revised manuscript should be accepted for the standard of journal.

Reviewer 3 Report

The authors' reply to the comments is understandable, and I figure out the novelty of this study; however, it seems that the revision is not consistent with the reply as follows:

  • The first paragraph in the reply: "... The relevant discussion has been added in the revised manuscript as follows."
    --> Where did the authors revise regarding this reply? Please describe.
  • The second paragraph in the reply: "...In addition, the LOD for the nanorods and network nanowires are 6.78 and 4.93 ppm, respectively. lower LOD value for the network nanowires indicate its effective detecting in practical application."
    --> Discussion on the LOD would be beneficial. Why not include that in the revised manuscript?

As long as the points above are revised, the manuscript would be judged to be acceptable.

Author Response

The authors' reply to the comments is understandable, and I figure out the novelty of this study; however, it seems that the revision is not consistent with the reply as follows:

The first paragraph in the reply: "... The relevant discussion has been added in the revised manuscript as follows."--> Where did the authors revise regarding this reply? Please describe.

We appreciate the reviewer for this comment. We are sorry that the we are not carful and missed it. The relevant discussion has been added in the revised manuscript as follows.

Herein, to evaluate the sensor response of gas sensor, the surface characteristics of gas-sensing material could be tailored via hybrid structures. This work reports a simple preparation, characterization and H2S gas sensing application of WO3 nanostructure based sensors. To compare the sensing performances of the prepared network nanowires, the rod-like structure was also synthesized and examined.

The second paragraph in the reply: "...In addition, the LOD for the nanorods and network nanowires are 6.78 and 4.93 ppm, respectively. lower LOD value for the network nanowires indicate its effective detecting in practical application."--> Discussion on the LOD would be beneficial. Why not include that in the revised manuscript? As long as the points above are revised, the manuscript would be judged to be acceptable.

We thank the reviewer for this comment. The relevant discussion has been added in the revised manuscript as follows.

 In addition, the LOD for the nanorods and network nanowires are 6.78 and 4.93 ppm, respectively. lower LOD value for the network nanowires indicates its effective detecting in practical application.